# Tea as a Source of Biologically Active Compounds in the Human Diet

**DOI:** 10.3390/molecules26051487

**Published:** 2021-03-09

**Authors:** Joanna Klepacka, Elżbieta Tońska, Ryszard Rafałowski, Marta Czarnowska-Kujawska, Barbara Opara

**Affiliations:** Department of Commodity Science and Food Analysis, The Faculty of Food Sciences, University of Warmia and Mazury in Olsztyn, 10-719 Olsztyn, Poland; elzbieta.tonska@uwm.edu.pl (E.T.); rych@uwm.edu.pl (R.R.); marta.czarnowska@uwm.edu.pl (M.C.-K.); barbara@opara.com.pl (B.O.)

**Keywords:** tea, phenolics, antioxidant activity, minerals, food safety, recommended daily intake

## Abstract

Due to the different levels of bioactive compounds in tea reported in the literature, the aim of this study was to determine whether commercially available leaf teas could be an important source of phenolics and selected minerals (copper, manganese, iron, zinc, magnesium, calcium, sodium, potassium) and if the differences in the content of these components between various types of tea are significant. It was found that both the amount of these compounds in tea and the antioxidant activity of tea infusions were largely determined by the origin of tea leaves as well as the processing method, which can modify the content of the studied components up to several hundred-fold. The group of green teas was the best source of phenolic compounds (110.73 mg/100 mL) and magnesium (1885 µg/100 mL) and was also characterised by the highest antioxidant activity (59.02%). This type of tea is a great contributor to the daily intake of the studied components. The average consumption of green tea infusions, assumed to be 3–4 cups (1 L) a day, provides the body with health-promoting polyphenol levels significantly exceeding the recommended daily dose. Moreover, drinking one litre of an unfermented tea infusion provides more than three times the recommended daily intake of manganese. Tea infusions can be a fairly adequate, but only a supplementary, source of potassium, zinc, magnesium, and copper in the diet. Moreover, it could be concluded that the antioxidant activity of all the analysed types of tea infusions results not only from the high content of phenolic compounds and manganese but is also related to the presence of magnesium and potassium.

## 1. Introduction

Tea (*Camelia sinensis*) is one of the most popular beverages consumed around the globe. It is cultivated in approx. 30 countries, the area of its cultivation has constantly been increasing, and this trend is expected to continue until 2027 [1,2]. The major tea producers are China, India, Kenya, and Sri Lanka as well as Turkey and Vietnam [1] and these economies export tea to many countries, including Poland.

For some consumers, the reason for consuming tea is its sensory quality, while for others, it is its nutritional value. These properties are primarily determined by the variety of tea leaves as well as the cultivation and harvesting conditions and the processing methods [3,4]. The most common classification of teas involves their division according to the leaf fermenting method. A distinction is made here between the almost unfermented white, yellow, and green teas, partially fermented oolong and pu-erh, and the fully fermented black tea [5]. Due to the sensory characteristics, the most popular is black tea, although interest in green tea has also been on the increase for many years [6].

The beneficial effect of tea on human health is primarily due to its high antioxidant activity, which results mainly from the presence of phenolic compounds, of which tannins and catechins, as well as their derivatives, are primarily found in tea [7]. The main polyphenols of fresh tea leaves are flavan-3-ols, whose content and structure change depending on technological processes. In green tea, catechins occur in monomeric forms and consist mainly of epicatechin, epigallocatechin, epicatechin gallate and epigallocatechin gallate [8]. During the process of semi-fermentation from green tea to oolong tea, catechins can be partially oxidised by polyphenol oxidases and peroxidases to dimers, such as theasinensins, theaflavins, and other high molecular weight polyphenols. In fully fermented black tea, many more catechins (compared to other types of tea) are transformed into theaflavins and thearubigins [9]. Phenolic compounds found in tea have an effect on the colour and aroma of its infusions and determine their characteristic bitter and astringent flavour [10]. 

The best-known and most widely described is the antioxidant activity of phenolic compounds and their stabilising effect on the capillary vessel walls [11,12]. Epidemiological studies have demonstrated a correlation between the consumption of phenolic compounds and a reduced incidence of certain types of neoplasms and other chronic diseases, e.g., cardiovascular, neurodegenerative diseases, and type II diabetes [6,9,13,14,15,16,17,18]. The biological effects of phenolic compounds also include the antiosteoporotic, antiatherosclerotic, antiallergic, anti-obesity, antifibrotic, hypolipidemic, hypocholesterolemic, antidiabetic, antiviral, antimutagenic, antimicrobial, and even antidepressant effects [19,20,21,22,23]. Ullah et al. [12] believe that due to the wide range of health-promoting effects of polyphenols, their conjugates in combination with important drugs may enhance the potency of these compounds and possibly even extend their therapeutic effects. The studies of Diniz et al. [24] are particularly interesting in this aspect because these authors showed that quercetin (which is a flavonoid, the main group of polyphenols) can be used as a tool in the treatment of renal dysfunction, which may prevent the clinical deterioration of patients infected with COVID-19 and, consequently, improve their health and reduce the mortality caused by this disease. 

Significant components of tea also include important minerals such as copper, manganese, iron, zinc, magnesium, calcium, sodium, and potassium [10]. Minerals are incorporated in all tissues of the body and are involved in many life processes by determining their correct course [25,26,27]. An excessively small or large amount of these minerals in the diet may lead to characteristic disorders and increase the risk of “diseases of affluence” (osteoporosis, hypertension, cancer, coronary heart disease, and diabetes) [27,28]. 

Various types of tea can be found on the food market. They differ primarily in the location and conditions of leaf cultivation and harvesting and processing methods. Moreover, tea leaves are sold in various degrees of fineness and as blends with different ingredients. Studies conducted in many countries indicate a diversity of different types of teas available for sale due to the content of polyphenols and minerals [10,29,30,31]. The nutritional importance of tea leaves and their contribution to the daily intake of these compounds has rarely been analysed, and most often, studies are based on a few selected types of tea. In the available literature, only one study of this type conducted on teas available on the Polish market was found and it only covered different kinds of green tea leaves [31]. 

Considering the above, the study aimed to determine to what extent the contents of phenolic compounds and minerals are modified by the type of tea available for sale and to evaluate the tea contribution to the daily intake of these components.

## 2. Materials and Methods

### 2.1. Food Samples

The study material comprised the leaves of 33 various types of tea without additives, purchased in June 2020 in a shop in Olsztyn (Warmia and Mazury region, Poland), belonging to a chain of shops specialising in the sale of this beverage. These were all teas with no other ingredients added, available at this store at the time of purchase.

The infusions were prepared by pouring 100 mL boiling deionised water (obtained in a Millipore Simplicity water purification system-Merck, Darmstadt, Germany) into 2.0 g of tea material. After 15 min of brewing under cover, infusions were filtered through paper filters (Eurochem BGD, Tarnów, Poland), and the obtained solutions samples were taken for further research.

### 2.2. Determination of Total Content of Phenolic Compounds

The total phenolics were determined spectrophotometrically according to Ribereau-Gayon method [32] in the modification by Guo et al. [33] using the addition of 0.5 mL of Folin-Ciocalteu reagent (Aktyn, Suchy Las, Poland) and 3 mL of 14% sodium carbonate (Stanlab, Poland) to tea infusions (0.04 mL) and measurement of the absorbance at a wavelength of 720 nm against the reference sample after 1 h (Thermo Scientific, Helios Zeta UV-VIS, Madison, WI, USA). The results were expressed as gallic acid equivalent with a reference curve plotted for this acid:(1)$$y\;=\;0.0223x\;-\;0.1916,\;R^{2}=\;0.990$$

### 2.3. Determination of Antioxidant Activity

The ability to neutralise the DPPH (2.2-diphenyl-1-picrylhydrazyl) radical was determined on the basis of colorimetric changes in the concentration of the stable DPPH radical in relation to the control sample [34,35]. The phenolic extract solution (0.3 mL) was mixed with a methanol (Stanlab, Lublin, Poland) solution containing DPPH (Aldrich, St. Louis, MO, USA) radicals (0.4 mM, 4 mL). Measurement of the absorbance was made at a wavelength λ = 517 nm, after 20 min incubation at room temperature, without light (Thermo Scientific, Helios Zeta UV-VIS, Madison, WI, USA). The ability of the tested extracts to counteract the oxidation reaction was calculated from the formula:(2)$$\%\;\mathrm{inhibition}\;=\;100\;-\;\left\{\frac{\left[\left(A_{w}\;-\;A_{0}\right)\;\times \;100\right]\;}{A_{k}}\right\}$$
where: A_w_—absorbance of a specific sample (the tested extract); A_0_—absorbance of the zero sample; A_k_—absorbance of the control sample (with a synthetic DPPH radical).

### 2.4. Determination of Copper (Cu), Manganese (Mn), Iron (Fe), Zinc (Zn), Magnesium (Mg) and Calcium (Ca)

The contents of individual minerals were determined by flame atomic absorption spectrometry (acetylene—air flame) using a Thermo iCE 3000 Series (Madison, WI, USA) atomic absorption spectrometer, with a Glite data station, background correction (deuterium lamp) and appropriate cathode lamps [36]. For the calcium determination, a 10% aqueous solution of lanthanum chloride was added to all measured solutions in a quantity, ensuring a final La^+3^ concentration of 1%. The selected elements determination was performed at the following wavelengths: 324.8 nm (Cu), 279.5 nm (Mn), 248.3 nm (Fe), 213.9 nm (Zn), 285.2 nm (Mg), and 422.7 nm (Ca). 

The method was validated by a simultaneous analysis of reference material (INCT-TL-1, tea leaves) with an accuracy for Cu, Mn, Fe, Zn, Mg, and Ca of 99.0%, 96.8%, 97.8%, 98.6%, 98.7%, and 99.1%, respectively. Reference materials were analysed in the form of water infusions prepared and assayed in a manner similar to that used for the studied tea leaves (tea weight to volume of boiling deionised water ratio was 1:50). The limit of detection for Cu, Mn, Zn, and Mg was 0.05 µg/mL, 0.2 µg/mL for Fe and 0.5 µg/mL for Ca. 

### 2.5. Determination of Sodium (Na) and Potassium (K)

The concentrations of sodium and potassium were determined by the emission technique (acetylene-air flame). The analyses were performed using an atomic absorption spectrometer Thermo iCE 3000 Series (Waltham, MA, USA), equipped with a Glite data station, operating in an emission system. The determination was carried out at the following wavelengths: 589.0 nm (Na) and 766.5 nm (K). 

The method was validated by a simultaneous analysis of reference material (INCT-TL-1, tea leaves) with an accuracy for Na and K of 101.6% and 98.8%, respectively. Reference materials were analysed in the form of water infusions prepared and assayed in a manner similar to that used for the studied tea leaves (tea weight to volume of boiling deionised water ratio was 1:50). The limit of detection for Na was 0.5 µg/mL and 2 µg/mL for K. 

### 2.6. Statistical Data Analysis

The data were analysed using the Statistica software package version 13.1 (StatSoft, Kraków, Poland, 2016). Significant differences were calculated according to an Anova analysis and Duncan’s Multiple range test. Differences at the 5% level were considered statistically significant. Correlation coefficients between analysed components were determined by Pearson’s correlation analysis at the *p* < 0.05 confidence level.

## 3. Results and Discussion

An analysis of the data provided in Table 1 reveals a great diversity among the tested teas in terms of the level of distinguishing features that indicate the nutritional value of their infusions. 

The highest level of phenolic compounds (152.12 mg/100 mL infusion) was found in a sample of green tea originating from India (No 13), while the lowest content of these compounds was determined in a black tea infusion originating from China (No 23) and pu-erh (No 22), in which they were found at the levels of 43.38 and 46.03 mg/100 mL, respectively. The variation in polyphenol levels in various tea infusions is also indicated inter alia by studies by Almeida et al. [37] and Samadi & Fard [29], who also demonstrated that the best source of these compounds were green tea leaves. 

Tea infusions indicated for the total phenolic content were also similarly distinguished from other samples in terms of antioxidant activity, measured by the amount of DPPH radical being neutralised. An infusion of green tea originating from India (No 13), whose ability to counteract the oxidation reaction was 89.76%, was the most active in this regard, while an infusion of black tea originating from China (No 23) exhibited an activity of this kind being almost five times lower (17.75%). Such a high antioxidant activity of green tea may be related to great amounts of catechins, which, according to Szajdek & Borowska [38,39], are 90% responsible for the total antioxidant capacity of this type of tea. A compound of a particular significance in this regard is epigallocatechin gallate, which contains, in its molecule, eight free OH groups that are responsible for its high antioxidant activity. In black and oolong tea, the dominant polyphenols include theaflavins and thearubigins, which are formed by biochemical transformations occurring in the leaf fermentation process, which results in a reduction in the catechin content by approx. 85%. Black tea, due to the presence of theaflavins, exhibits poorer antioxidant properties as compared to green teas, which contain catechins [38,39,40].

The differences in the antioxidant activity of green and black teas were also confirmed by the data summarised in Table 2, which present the mean values of the examined discriminants determined for individual types of tea. Based on the data, it can be concluded that the best source of phenolic compounds (110.73 mg/100 mL) and magnesium (1885 µg/100 mL) was the group of green teas, which was also characterised by the highest antioxidant activity (59.02%). The elevated level of total phenolic compound and magnesium contents, which accompanies high antioxidant activity, indicates the particular antioxidant effect of these compounds. This relationship is confirmed by the values of correlation coefficients specified between them and presented in Table 3. They indicate statistically significant relationships between the antioxidant activity and both the total phenolic compound level (r = 0.9288) and magnesium content (0.7472) and between the phenolic compound content and magnesium content (r = 0.7160). This may be related to the occurrence of chemical bonds between the phenolic compounds and magnesium, which is indicated in a study by Ghosh et al. [41]. They demonstrated that quercetin (which is an important phenolic component of tea) can react with a magnesium cation (Mg^+2^) through the chelation site in the quercetin molecule, which increases its antioxidant activity. Many authors proved that polyphenols are the most important antioxidant components found in tea leaves [3,5,6,10,19]. Samadi & Fard [29] indicated that a particular antioxidant effect in tea infusions was exhibited by tannins and flavonoids, for which the correlation coefficient values specifying their relationships with antioxidant activity were statistically significant and amounted to r = 0.721 and r = 0.740, respectively. Nordin et al. [42] also emphasise the significance of the method of measuring the antioxidant activity of tea infusions in determining its relationship with the level of phenolic compounds. They demonstrated that measuring it using ferric-reducing antioxidant power and 2.2-diphenyl-1-picrylhydrazyl assays (which was applied in this study) has a significant effect on the value of the determined correlation coefficients which, as regards the relationship between them and the polyphenol level, amounted to 0.9772 and 0.8705, respectively.

The best source of magnesium (3688 µg/100 mL) was green tea originating from India (No 13), and it was also followed, in terms of this element content, by infusions of green tea cultivated in China (No 5, 9, and 11), which contained this compound at levels ranging from 2419 to 2576 µg/100 mL (Table 1). The high magnesium content noted for certain green tea infusions may be due to the fact that this element plays a crucial role in the structure of chlorophyll, which changes rather easily under the influence of the tea leaf processing [43,44]. The rate of these changes is particularly determined by the duration and temperature of the process, which can decrease or increase the magnesium level in tea leaves. Not only does it have an effect on differences in this element content, noted for various types of tea, but it can also produce differences between teas of the same type, as was noted by Zaguła et al. [8].

A very high variation in tea infusions due to their copper content was found (Table 1). Two infusions of green tea originating from China, in which this compound was found at a level of 9.89 µg/100 mL (tea No 10) and 9.40 µg/100 mL (tea No 5), were characterised by the highest level of this element. An infusion of tea originating from Japan, which also belonged to the green tea group, was characterised by an almost 30-fold lower content of this element (0.37 µg/100 mL, sample No 17). Most of the literature data do not indicate such a large differentiation of the teas available on the market in terms of copper levels, stating that these differences may be 2-fold [10] or 3-fold [29,31], although some studies conducted on teas available in Poland indicate that these differences can be 20-fold [30]. Qin & Chen [45] reported that the great variation in tea leaves available on the market in terms of the copper content may result, besides the conditions prevailing during their cultivation and harvesting, from contamination with this element due to the contact of leaves with various surfaces used during the production of tea.

Even greater differentiation of the analysed infusions was found in terms of sodium content (Table 1). White tea (No 2), which is the best source of this element, contained it at a level of 737 µg/100 mL, and one of the green teas (No 12), the poorest source of this compound, contained it at an almost 200-times lower level. Some authors have noted that these differences can be as much as 400-fold, adding that such a great variation in the tested tea infusions in terms of the sodium level may primarily result from the differences in the leaf cultivation method and the type of soil in which they are cultivated (in particular its salinity) [30,46].

The large diversity of the analysed tea infusions in terms of the content of copper and sodium was not confirmed by the average values of these elements, calculated for individual types of tea, as presented in Table 2. The standard deviation determined for these values indicates a large variation in infusions within individual groups, which may result in smaller differences between the individual groups of tea detected in statistical tests.

The analysed tea infusions varied the least in terms of the potassium content (Table 1 and Table 2). Its highest amounts were found in a green tea originating from India (No 13), whose infusion contained this component at a level of 45,535 µg/100 mL, while the infusions of one of the green teas (No 7) and oolong (No 20), characterised by the lowest potassium content, contained it at an almost 3-fold lower level (Table 1). A similar level of potassium found in green tea infusions was also indicated by Zaguła et al. [8], who reported that, depending on the method for preparing infusions and the origin of tea, the content of this element ranges from 45,000 to 55,100 µg/100 mL.

Pu-erh teas, which are a very poor source of most of the compounds discussed above, were the best (of all the tested infusions) source of iron and zinc (Table 1 and Table 2). The iron level determined in them (13.32 µg/100 mL) was over 7-fold higher than that in white and yellow tea infusions, and the zinc level (86.7 µg/100 mL) exceeded the content of this element determined in the oolong tea infusion by almost 10 times (Table 2). A much higher (several times) content of iron and zinc in the leaves of pu-erh tea, as compared to other tea types, was also indicated by Garbowska et al. [30] and Lv et al. [47], who recommend selecting this kind of tea, particularly for people prone to deficiencies in these elements.

The variation in the level of minerals in various tea infusions has been indicated by numerous authors [7,10,25,26,28,29,30,31] who, besides the tea production method, also noted other factors impacting this variation such as the variety, location and method of leaf cultivation and the duration and technique of harvesting, storage and transport. The differences in the content of minerals shown by the authors depend to a large extent on the type of tea leaves available in the country, and the differences between them may be several- [10,29], several dozen- [31] or even several hundred-fold [30]. Zhang et al. [48] believe that such a high variation in the mineral content in tea leaves primarily results from the location of their cultivation. They analysed fresh leaves of 87 teas cultivated in three various regions of China and demonstrated that the differences in the contents of 11 elements between them were so great that it enabled the development of a mathematical model of the relationship between the location of tea plant cultivation and the mineral content in the plants. Konieczyński et al. [10], by confirming the variation in different types of teas due to the mineral content as well as the levels of polyphenols and their antioxidant properties, indicated that these distinguishing features can be applied to mathematically distinguish between various kinds of tea available on the market.

An analysis of the values of correlation coefficients presented in Table 3 confirms the antioxidant significance of phenolic compounds, which was discussed above. The value of the antioxidant potential of tea infusions is also affected by the levels of magnesium (r = 0.7472), potassium (r = 0.5158), and manganese (r = 0.4587), which means that the antioxidant effect of infusions increases with an increase in these elements. Significant correlation relationships between the magnesium and manganese contents and the polyphenol level were also indicated by Konieczyński et al. [10], who stated that they are largely determined by the type of phenolic compounds found in tea. The antioxidant effects of manganese, which is a micronutrient being part of antioxidant enzymes, such as glutathione peroxidase or superoxide dismutase, is well-known and documented in the literature [11,49]. There are fewer studies concerning the effects of magnesium and potassium on the antioxidative potential of various food raw materials. However, Ghosh et al. [41] reported that magnesium found in tea had an effect on the antioxidant action of certain phenolic compounds. The antioxidative action of this element in herbs was reported by Narendhran et al. [50], while Hosseini et al. [51] noted the effect of magnesium on the antioxidant potential of various types of food and its role in preventing obesity. The positive effect of potassium on the antioxidant activity of wheat is reported in two studies by Ahanger & Agarwal [52,53], while Taha et al. [54] found a positive effect of this element on the antioxidant properties of growing soybean.

An analysis of the correlation coefficient values also indicates the occurrence of statistically significant relationships between multiple minerals, of which the most significant ones appear to be the positive correlation relationships between micronutrients which are antioxidant enzymes, i.e., between manganese and copper, manganese and zinc, and zinc and iron, which may be indicative of the similarity of the chemical bonds in which they are found in tea leaves.

Due to the scale of the daily demand and their content in the body, minerals are divided into two groups: macronutrients, i.e., elements for which the required daily demand is higher than 100 mg/person and their content in living organisms is at a level above 0.01%, and micronutrients (trace elements), for which the required daily demand is at a level lower than 100 mg/person, and the content in organisms is below 0.01% [26,27]. The data summarised in Table 1, indicate that, for macronutrients, the tested tea infusions were the best source of potassium which, depending on the type and origin of tea leaves, ranged from 15,506 to 45,535 µg/100 mL infusion. In terms of content, it was followed by magnesium found in amounts ranging from 405 to 3688 µg/100 mL, along with sodium (from 4.0 to 737 µg/100 mL) and calcium (44.6–445 µg/100 mL). What is noteworthy is the particularly high content of manganese, classified as a micronutrient, which was found in the analysed tea infusions in an amount greater than that for sodium and calcium, and its content ranged from 185 to 1329 µg/100 mL. Apart from manganese, tea infusions were also a source of other micronutrients found in much lesser amounts, i.e., zinc, iron, and copper, whose content ranged from 7.8 to 149 µg/100 mL, from 1.68 to 14.46 µg/100 mL, and from 0.37 to 9.89 µg/100 mL, respectively. 

Many factors have an effect on the body’s demand for minerals, including e.g., age, sex and physical activity [27,28]. Such a relationship is not determined for phenolic compounds, because no daily demand is determined for them, with their consumption being regarded as a recommended nutritional pattern. Del Bo’ et al. [14] analysed literature data published over a period of 10 years, which determined the relationship between the average consumption of these compounds in an average diet followed in various parts of the world and the incidence of specific diseases. Their study demonstrated that a significantly reduced susceptibility of various illnesses was observed when polyphenols were consumed at a level above 500 mg and, in certain cases, 900 mg a day, because such doses determine the observation of a positive biological effect of their action. Based on the data provided in Table 2, it should be concluded that with an average intake of all analysed tea infusions, assumed to be 3–4 cups (1 L) on a daily basis, it is possible to obtain a positive health-promoting effect of phenolic compounds due to the provision of the lower of the recommended levels of these compounds (500 mg). Drinking green teas can provide the body with approx. 1107.3 mg of these components, which provides an amount considerably exceeding both levels of these compounds recommended for consumption. 

Table 4 presents the recommended daily intake of the determined minerals as well as the percentage in which it is satisfied by the average tea consumption, assumed to be one litre a day. Such an amount of infusion allows the demand for manganese (for drinking white and green tea, is satisfied from 219% to 341% for children and youth, respectively, and from 210% to 284% for adults), to be supplemented to the greatest extent compared with other elements. Drinking other types of tea also provides the body with large amounts of this element, as it satisfies the daily demand for this component from 95% to 211%. These are noteworthy amounts due to the importance of manganese in the human diet. It is an essential element for the proper metabolism, as it forms catalytic centres in many enzymes and actively participates in the activation of enzymes that catalyse the transformations of proteins, lipids, and carbohydrates and in the connective tissue and bones formation process [26]. It also plays an important role in the process of body growth and reproduction processes and stabilises blood sugar levels. Chronic manganese deficiency may lead to bone deformation, stunted growth, motor coordination disorders, diabetes or even schizophrenia, toxicity and neurological disorders [26,28,55]. Such a high level of manganese in tea does not seem to pose the risk of its excessive consumption, provided that the amount of tea infusions consumed daily does not significantly exceed one litre. Jarosz [27] reports that no adverse effects of this element consumed in the diet at the level of 8–9 mg per day have been observed in humans.

Drinking one litre of infusions of most of the analysed tea types provides the body with approx. 10% of the daily potassium dose recommended for consumption, with oolong tea being an exception in this regard, as it provides half the amount of this element. 

The analysed white, yellow, and black teas consumed in an amount of one litre a day provide from 5.56% to 7.14% of the recommended daily intake of copper for children and youth and 5.56% for adults. A slightly lower amount of this element is provided by green and oolong tea infusions (from 4.44% to 5.71% of the recommended daily intake) and pu-erh infusions (from 3.33% to 4.29%).

The consumption of pu-erh tea can be a good supplement to the recommended daily intake of zinc, as one litre of infusion of this tea provides from 7.91% to 10.90% of this amount. Other tea infusions satisfy the daily demand for this element in amounts several times lower, and oolong tea infusions do it to a 10 times lower degree. 

The consumption of green tea can also be noteworthy in terms of magnesium, since 1000 mL of this tea provides from 4.61% to 7.88% of the daily amount of this element recommended for children and youth and from 4.50% to 6.10% of the level recommended for adults. The poorest source of this element are oolong tea infusions, which satisfy only 1–2% of the recommended intake of this component. 

Irrespective of the tea production method, its average consumption does not provide significant amounts of iron, calcium, and sodium, as the consumption of one litre of tea on a daily basis satisfies the recommended daily intake of these elements at the following levels, respectively: 0.11–1.30%, 0.06–0.43% and 0.04–0.36%. 

Many authors confirm the significant role of tea infusions, demonstrated in this study, in providing the amount of manganese recommended for daily intake [25,30], while Brzezicha-Cirocka et al. [31] also indicate that they may provide the body with a noticeable level of copper, magnesium, and zinc, and only a very small amount of calcium and sodium.

## 4. Conclusions

The high variety of teas available on the market in terms of phenolics and minerals (as defined in the study) indicates that health-conscious consumers should pay special attention to the appropriate selection of this product because the differences in the polyphenol content may be four–fold and for the mineral content even several hundred–fold. The best source of these components are unfermented or partially fermented teas in contrast to fully fermented black teas. The average consumption of green tea infusions, assumed to be 3–4 cups (1 L) a day, may be a significant source of phenolic compounds and manganese in the diet, because it provides the body with an amount of these components that significantly exceeds the levels recommended for daily consumption. 

This study also confirmed that the antioxidant effect of tea infusions results mainly from the presence of phenolic compounds and manganese. However, this antioxidant activity is also associated with the occurrence of magnesium and potassium, which has, to date, been demonstrated mainly in relation to food products other than tea.

Due to the large differences in the content of mineral compounds indicated in this study, research in this area should be continued in order to create a mathematical model of the relationship between the content of these compounds determined in tea leaves available in a particular country over the years, and the method of their production and the place of origin. Future studies should also focus on verifying the health-promoting effect of various tea infusions on living organisms because the biological effect of polyphenols and minerals results, among others, from their quantity, bioavailability, and time of consumption.

## Figures and Tables

**Table 1 molecules-26-01487-t001:** The content of selected biologically active components determined in different tea infusions.

No	Kind of Tea	Country	Province	Total Phenolics (mg/100 mL)	Antioxidant Activity (%)	The Content of Minerals (µg/100 mL)							
						Cu	Mn	Fe	Zn	Mg	Ca	Na	K
1.	white	China	Fujian	65.05 ± 1.40 p	37.27 ± 3.54 jkl	4.75 ± 0.12 g	318 ± 0.79 m	1.68 ± 0.09 n	32.7 ± 1.57 hi	1036 ± 13.80 t	85.6 ± 0.43 o	198 ± 2.23 d	21,485 ± 83.5 s
2.		China	Yunnan	58.21 ± 0.75 q	31.89 ± 1.36 mn	5.88 ± 0.34 f	645 ± 3.19 d	2.31 ± 0.28 lmn	37.2 ± 1.05 f	1230 ± 0.58 r	71.8 ± 0.61 opqr	737 ± 15.30 a	33,452 ± 109 f
3.	yellow	China	Anhui	79.10 ± 1.74 mn	40.51 ± 1.47 ij	4.75 ± 0.17 g	289 ± 0.97 n	1.96 ± 0.20 mn	43.4 ± 3.42 e	1752 ± 7.02 n	141 ± 1.74 l	74.1 ± 1.41 jk	29,923 ± 142 j
4.	green	China	Anhwei	90.46 ± 0.95 j	49.41 ± 3.69 fg	7.59 ± 0.41 d	324 ± 4.48 m	9.14 ± 0.52 c	31.3 ± 1.56 ij	1494±6.43 o	307 ± 8.01 e	103 ± 1.04 h	26,240 ± 99.9 o
5.		China	-	107.88 ± 0.34 g	65.09 ± 1.22 c	9.40 ± 0.20 b	621 ± 4.93 e	5.93 ± 0.29 efg	63.3 ± 1.08 b	2576 ± 5.29 b	337 ± 35.60 d	60.5 ± 0.49 no	33,056 ± 102 gh
6.		China	-	83.21 ±0.72 l	43.51 ± 0.60 hi	2.44 ± 0.10 jk	298 ± 2.74 n	12.20 ± 0.68 b	14.1 ± 1.59 q	492 ± 4.00 x	366 ± 1.50 b	21.2 ± 0.26 u	21,127 ± 60.9 t
7.		China	-	76.23 ± 3.63 no	35.28 ± 1.79 klm	1.96 ± 0.06 l	207 ± 1.19 s	11.83 ± 1.07 b	7.8 ± 0.13 r	405 ± 2.65 y	445 ± 2.25 a	10.1 ± 0.87 v	15,506 ± 96.6 v
8.		China	Jiangs	117.26 ± 1.06 f	70.61 ± 0.89 b	8.00 ± 0.27 c	1329 ± 9.32 a	6.83 ± 1.03 de	62.2 ± 1.74 b	2386 ± 10.30 e	226 ± 22.00 i	56.6 ± 0.44 op	39,077 ± 176 b
9.		China	Yunnan	128.17 ± 0.68 d	72.52 ± 1.58 b	3.77 ± 0.17 h	450 ± 2.18 h	4.48 ± 0.21 jk	35.1 ± 0.45 fgh	2528 ± 11.80 c	164 ± 1.89 k	39.5 ± 0.14 s	32,580 ± 109 i
10.		China	Yunnan	124.51 ± 0.96 e	60.80 ± 3.78 cd	9.89 ± 0.45 a	1130 ± 9.00 b	6.64 ± 0.48 def	46.8 ± 2.25 d	2092 ± 15.70 j	58,4 ± 1.31 r	41.0 ± 0.17 rs	36,430 ± 186 d
11.		China	Zhejiang	99.13 ± 2.97 hi	57.30 ± 086 de	0.66 ± 0.08 n	790 ± 7.30 c	9.03 ± 1.40 c	35.8 ± 2.74 fg	2419 ± 4.00 d	161 ± 1.34 k	41.8 ± 0.32 rs	38,706 ± 143 c
12.		China	Zhejiang	86.69 ± 0.28 k	49.52 ± 0.68 g	2.66 ± 0.05 jk	496 ± 3.81 g	4.83 ± 0.18 ij	21.6 ± 0.78 nop	1429 ± 3.21 p	182 ± 1.40 j	4.0 ± 0.25 w	23,608 ± 59.4 q
13.		India	Assam	152.12 ± 1.35 a	89.76 ± 1.26 a	2.60 ± 0.11 jk	555 ± 2.26 f	7.01 ± 0.28 d	52.2 ± 2.46 c	3688 ± 28.00 a	77,8 ± 0.18 op	27.0 ± 0.35 tu	45,535 ± 436 a
14.		Japan	-	96.55 ± 0.79 i	54.16 ± 1.60 ef	1.23 ± 0.07 m	397 ± 3.80 k	8.34 ± 0.42 c	15.5 ± 1.23 q	1246 ± 6.51 r	155 ± 0.49 k	72.3 ± 0.75 jkl	23,896 ± 107 q
15.		Japan	-	97.90 ± 0.87 i	33.78 ± 4.97 lm	1.21 ± 0.10 m	332 ± 0.85 l	5.41 ± 0.19 ghi	26.9 ± 0.25 kl	1237 ± 4.00 r	268 ± 8.81 fg	138 ± 0.94 f	21,396 ± 46.4 st
16.		Japan	Miyazaki	102.16 ± 0.9h	59.20 ± 0.40 d	1.83 ± 0.10 l	428 ± 7.25 j	5.79 ± 0.33 fgh	32.1 ± 2.13 i	2014 ± 20.1 k	122 ± 2.30 m	77.7 ± 5.80 j	33,097 ± 98.9 gh
17.		Japan	Shizouka	144.54 ± 1.66 b	57.40 ± 5.32 de	0.37 ± 0.05 n	290 ± 4.04 n	3.83 ± 0.16 k	22.8 ± 0.20 mno	1983 ± 6.40 l	64.9 ± 1.11 pqr	64.1 ± 0.46 mn	22,604 ± 51.8 r
18.		Korea	Jeju	128.58 ± 0.91 d	70.82 ± 3.35 b	1.35 ± 0.08 m	316 ± 0.70 m	8.91 ± 0.85 c	28.4 ± 0.82 k	2216 ± 12.90 h	68.1 ± 0.82 pqr	226 ± 11.60 c	30,193 ± 118 j
19.		Vietnam	-	136.24 ± 1.37 c	75.09 ± 3.64 b	3.03 ± 0.21 i	236 ± 19.7 q	5.81 ± 0.30 fgh	33.7 ± 2.80 ghi	1957 ± 7.37 m	73.4 ± 0.43 opq	37.1 ± 1.13 s	28,007 ± 98.0 l
20.	oolong	China	-	62.66 ± 1.01 p	33.70 ± 1.06 lm	3.66 ± 0.15 h	317 ± 0.62 m	4.40 ± 0.37 jk	8.8 ± 0.41 r	496 ± 4.51 x	433 ± 4.08 a	83.9 ± 0.28 i	16,752 ± 40.9 u
21.	pu-erh	China	Yunnan	51.23 ± 6.18 r	21.46 ± 2.41 qr	3.12 ± 0.03 i	224 ± 0.12 r	12.17 ± 0.52 b	23.7 ± 1.64 mn	1003 ± 12.80 u	265 ± 12.00 fg	226 ± 1.66 c	25,527 ± 517 p
22.		China	Yunnan	46.03 ± 1.29 st	20.79 ± 1.44 qr	2.37 ± 0.10 k	215 ± 1.45 s	14.46 ± 0.26 a	149 ± 1.81 a	997 ± 14.5 u	243 ± 5.26 h	292 ± 2.00 b	25,238 ± 210 p
23.	black	China	-	43.38 ± 3.76 t	17.75 ± 2.08 r	3.74 ± 0.17 h	223 ± 0.64 r	6.74 ± 0.16 de	24.5 ± 1.22 lm	1142 ± 9.45 s	44.6 ± 1.00 s	46.4 ± 0.43 qr	26,138 ± 33.4 o
24.		China	Yunnan	55.37 ± 0.75 q	28.59 ± 8.71 no	5.76 ± 0.10 f	438 ± 3.96 i	3.91 ± 0.17 k	31.2 ± 0.26 ij	1446 ± 9.85 p	61.6 ± 0.46 qr	30.6 ± 0.07 t	28,485 ± 95.7 k
25.		China	Yunnan	80.45 ± 0.62 lm	33.14 ± 1.21 lmn	5.78 ± 0.11 f	450 ± 1.26 h	5.99 ± 0.45 efg	20.9 ± 0.37 op	1736 ± 18.1 n	86.3 ± 2.67 o	103 ± 0.96 h	32,972 ± 37.3 h
26.		India	Assam	89.79 ± 1.36 jk	46.87 ± 2.37 gh	3.49 ± 0.12 h	253 ± 0.80 p	2.69 ± 0.12 lm	22.0 ± 0.10 mno	2309 ± 39.5 f	104 ± 4.47 n	84.0 ± 0.45 i	35,521 ± 649 e
27.		India	Assam	82.73 ± 0.42 l	35.97 ± 1.35 jklm	2.71 ± 0.21 j	185 ± 1.21 t	2.63 ± 0.31 lm	19.1 ± 1.32 p	1999 ± 3.79 kl	168 ± 0.26 k	55.4 ± 0.44 op	36,215 ± 131 d
28.		India	Assam	75.59 ± 2.5 o	37.92 ± 2.49 jkl	3.63 ± 0.14 h	213 ± 1.78 s	3.01 ± 0.24 l	20.7 ± 2.27 op	2242 ± 4.58 g	133 ± 1.68 lm	42.3 ± 0.26 rs	32,304 ± 202 i
29.		India	Ceylon	48.35 ± 0.39 rs	22.61 ± 1.56 pqr	6.28 ± 0.03 e	248 ± 1.73 p	4.56 ± 0.22 ijk	14.5 ± 0.26 q	963 ± 14.0 v	277 ± 3.69 f	160 ± 0.65 e	22,711 ± 16.9 r
30.		India	Darjeeling	93.00 ± 0.92 j	49.48 ± 1.69 fg	8.04 ± 0.17 c	253 ± 0.46 p	4.84 ± 0.07 ij	28.4 ± 0.26 k	2126 ± 10.4 i	73.4 ± 1.52 opqr	51.6 ± 0.52 pq	33,348 ± 57.6 fg
31.		India	mix 1*	61.84 ± 3.84 p	26.49 ± 1.84 op	6.34 ± 0.12 e	270 ± 1.05 o	4.96 ± 0.38 hij	29.4 ± 1.05 jk	1350 ± 4.58 q	255 ± 2.88 gh	116 ± 0.85 g	27,336 ± 67.6 m
32.		India	mix 2 *	55.56 ± 1.01 q	24.93 ± 1.63 opq	4.71 ± 0.20 g	320 ± 2.45 m	2.06 ± 0.12 mn	9.3 ± 0.40 r	870 ± 10.1 w	350 ± 3.53 c	70.2 ± 0.38 klm	26,608 ± 48.3 n
33.		Kenya	-	76.90 ± 0.51 no	39.48 ± 1.15 ijk	3.67 ± 0.08 h	292 ± 1.38 n	6.52 ± 0.32 def	21.4 ± 0.68 nop	2077 ± 3.00 j	101 ± 1.12 n	66.9 ± 2.16 lm	32,340 ± 52.8 i

* mix 1: Assam and Ceylon; * mix 2: Assam, Yunnan, and Lapsang; a, b, c–y—values in columns denoted by the same letters are not statistically different in the analysed type of tea at *p* < 0.05; Data are expressed as mean values ± standard deviations (SDs) of three samples (*n* = 3). The significance of differences was determined using Duncan’s test.

**Table 2 molecules-26-01487-t002:** The average values of selected biologically active components determined in particular types of tea infusions.

No	Kind of Tea	Total Phenolics [mg/100 mL]	Antioxidant Activity [%]	The Content of Minerals [µg/100 mL]							
				Cu	Mn	Fe	Zn	Mg	Ca	Na	K
1.	white	61.63 ± 3.88 bc	34.58 ± 3.80 bc	5.32 ± 0.66 a	482 ± 179.0 a	1.99 ± 0.39 d	34.9 ± 2.73 c	1133 ± 106 b	78.7 ± 7.62 d	468 ± 296 ab	27,469 ± 6555 a
2.	yellow	79.10 ± 1.7 4b	40.51 ± 1.47 b	4.75 ± 0.17 a	289 ± 0.971 a	1.96 ± 0.20 d	43.4 ± 3.42 b	1752 ± 7.02 a	141 ± 1.74 c	74.1 ± 1.41 d	29,923 ± 124 a
3.	green	110.73 ± 22.46 a	59.02 ± 14.80 a	3.62 ± 3.13 abc	512 ± 313.1 a	7.25 ± 2.47 b	33.1 ± 15.91 bcd	1885 ± 806 ab	192 ± 63.7 bc	63.7 ± 54.06 cd	29,441 ± 7891 a
4.	oolong	62.66 ± 1.01 bc	33.70 ± 1.06 bc	3.66 ± 0.15 b	317 ± 0.624 a	4.40 ± 0.37 c	8.80 ± 0.41 e	496 ± 4.51 c	433 ± 4.08 a	83.9 ± 0.28 c	16,752 ± 40.9b
5.	pu-erh	48.63 ± 4.90 c	21.12 ± 1.81 c	2.74 ± 0.41 c	219 ± 5.21 a	13.32 ± 1.31 a	86.7 ± 68.98 abcd	1000 ± 12.69 b	254 ± 14.44 b	259 ± 36.22 b	25,382 ± 387 a
6.	black	69.36 ± 16.66 bc	33.02 ± 10.06 bc	4.92 ± 1.59 a	286 ± 83.5 a	4.36 ± 1.60 bc	21.9 ± 6.28 d	1660 ± 512 ab	150 ± 97.47 bcd	75.1 ± 37.08 cd	30,362 ± 4196 a

a, b, c, d-values in columns denoted by the same letters are not statistically different in the analysed type of tea at *p* < 0.05. Data are expressed as mean values ± standard deviations (SDs). The significance of differences was determined using Duncan’s test.

**Table 3 molecules-26-01487-t003:** Correlations between analysed discriminants.

	TP *	AA *	Cu	Mn	Fe	Zn	Mg	Ca	Na
AA	0.9288 *								
Cu	−0.1114	−0.0210							
Mn	0.3813 *	0.4587 *	0.4328 *						
Fe	−0.0044	0.0167	−0.2298 *	0.0048					
Zn	0.0430	0.1064	0.1491	0.2555 *	0.3534 *				
Mg	0.7160 *	0.7472 *	0.1068	0.3859 *	−0.2084 *	0.2164 *			
Ca	−0.3056 *	−0.2755 *	0.0192	−0.1283	0.3433 *	−0.0800	−0.5425 *		
Na	−0.3183 *	−0.2870 *	0.0635	0.0117	−0.0341	0.2475 *	−0.2268 *	−0.1160	
K	0.4337 *	0.5158 *	0.2618 *	0.5185 *	−0.1839	0.2638 *	0.8652 *	−0.5287 *	−0.0152

* TP—total phenolic compounds; AA—antioxidant activity; numbers denoted by * and bold type indicate correlation coefficients significant at *p* ≤ 0.05. Correlation coefficients were estimated for *n* = 33 cases.

**Table 4 molecules-26-01487-t004:** Coverage of the demand for selected minerals after consuming 1 L (1000 mL) of tea infusions.

Type of Tea												
Average Content of Minerals Determined in Tea Groups, RDA *, DDC * and AI *	White		Yellow		Green		Oolong		Pu-Erh		Black	
	Children and Youth 10–18 Years Old	Adults 19–75 Years Old	Children and Youth 10–18 Years Old	Adults 19–75 Years Old	Children and Youth 10–18 Years Old	Adults 19–75 Years Old	Children and Youth 10–18 Years Old	Adults 19–75 Years Old	Children and Youth 10–18 Years Old	Adults 19–75 Years Old	Children and Youth 10–18 Years Old	Adults 19–75 Years Old
Cu (mg/L)	0.05	0.05	0.05	0.05	0.04	0.04	0.04	0.04	0.03	0.03	0.05	0.05
RDA	0.70–0.90	0.90	0.70–0.90	0.90	0.70–0.90	0.90	0.70–0.90	0.90	0.70–0.90	0.90	0.70–0.90	0.90
DDC (%)	5.56–7.14	5.56	5.56–7.14	5.56	4.44–5.71	4.44	4.44–5.71	4.44	3.33–4.29	3.33	5.56–7.14	5.56
Mn (mg/L)	4.82	4.82	2.89	2.89	5.12	5.12	3.17	3.17	2.19	2.19	2.86	2.86
AI	1.5–2.2	1.8–2.3	1.5–2.2	1.8–2.3	1.5–2.2	1.8–2.3	1.5–2.2	1.8–2.3	1.5–2.2	1.8–2.3	1.5–2.2	1.8–2.3
DDC (%)	219–321	210–268	131–193	126–161	233–341	223–284	144–211	138–176	100–146	95–122	130–191	124–159
Fe (mg/L)	0.02	0.02	0.02	0.02	0.07	0.07	0.04	0.04	0.13	0.13	0.04	0.04
RDA	10–15	10–18	10–15	10–18	10–15	10–18	10–15	10–18	10–15	10–18	10–15	10–18
DDC (%)	0.13–0.20	0.11–0.20	0.13–0.20	0.11–0.20	0.47–0.70	0.39–0.70	0.27–0.40	0.22–0.40	0.87–1.30	0.72–1.30	0.27–0.40	0.22–0.40
Zn (mg/L)	0.35	0.35	0.43	0.43	0.33	0.33	0.09	0.09	0.87	0.87	0.22	0.22
RDA	8–11	8–11	8–11	8–11	8–11	8–11	8–11	8–11	8–11	8–11	8–11	8–11
DDC (%)	3.18–4.38	3.18–4.38	3.91–5.38	3.91–5.38	3.00–4.13	3.00–4.13	0.82–1.13	0.82–1.13	7.91–10.9	7.91–10.9	2.00–2.75	2.00–2.75
Mg (mg/L)	11.3	11.3	17.5	17.5	18.9	18.9	5.00	5.00	10.0	10.0	16.6	16.6
RDA	240–410	310–420	240–410	310–420	240–410	310–420	240–410	310–420	240–410	310–420	240–410	310–420
DDC (%)	2.76–4.71	2.69–3.65	4.27–7.29	4.17–5.65	4.61–7.88	4.50–6.10	1.22–2.08	1.19–1.61	2.44–4.17	2.38–3.23	4.05–6.92	3.95–5.35
Ca(mg/L)	0.79	0.79	1.41	1.41	1.92	1.92	4.33	4.33	2.54	2.54	1.50	1.50
RDA	1300	1000–1200	1300	1000–1200	1300	1000–1200	1300	1000–1200	1300	1000–1200	1300	1000–1200
DDC (%)	0.06	0.07–0.08	0.11	0.12–0.14	0.15	0.16–0.19	0.29	0.36–0.43	0.20	0.21–0.25	0.12	0.13–0.15
Na (mg/L)	4.68	4.68	0.74	0.74	0.64	0.64	0.84	0.84	2.59	2.59	0.75	0.75
AI	1300–1500	1300–1500	1300–1500	1300–1500	1300–1500	1300–1500	1300–1500	1300–1500	1300–1500	1300–1500	1300–1500	1300–1500
DDC (%)	0.31–0.36	0.31–0.36	0.05–0.06	0.05–0.06	0.04–0.05	0.04–0.05	0.056–0.064	0.056–0.064	0.17–0.20	0.17–0.20	0.05–0.06	0.05–0.06
K (mg/L)	275	275	299	299	294	294	168	168	254	254	304	304
AI	2400–3500	3500	2400–3500	3500	2400–3500	3500	2400–3500	3500	2400–3500	3500	2400–3500	3500
DDC (%)	7.86–11.5	7.86	8.54–12.5	8.54	8.40—12.3	8.40	4.80–7.00	4.80	7.26–10.6	7.26	8.69–12.7	8.69

* RDA—Recommended Daily Allowance [mg/person]; AI—Adequate Intake [mg/person]; DDC—Daily demand coverage. The RDA and AI values come from the nutritional standards developed for the Polish population by Jarosz 2017 [27], while the DDC values were calculated based on the data presented in Table 2.

## Data Availability

Data is contained within the article.

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
