# Peer review of "Tea as a Source of Biologically Active Compounds in the Human Diet"

_molecules, 2021, doi:10.3390/molecules26051487_

Round 1

Reviewer 1 Report

The study presented by Klepacka and colleagues aimed to characterize phenolic compounds and selected minerals in 33 teas from 6 different kinds. The authors demonstrated differences between 33 different teas and differences between the six different kinds of tea. The authors also correlated the presence of phenolic compounds and Mn with high antioxidant activity in green tea. The study appeared to be very well conducted, with robust analysis and well discussed. I accept the paper in the present form. In the future, I would like to see the treatment of research animals with selected teas, confirming the antioxidant capacity of green tea in different organs, including the brain.

Reviewer 2 Report

This study is relatively comprehensive, but there are many scientific/technical items for which revisions are recommended.

The abstract section should be revised by inserting some key results, while keeping only the relevant information/data.

The Introduction hints the reader from a general subject area to a particular topic of inquiry, giving some background information and set the scope, context and significance of the research. In this case, although the authors mention 28 references, this section is of low quality. In this regard, the authors should rewrite/ revise the introduction section (literature review) in order to present the (recent) state of the art in the field of manuscript. Moreover, the novelty/ originality of the paper is not enough, adequately presented.

Some elements/ minerals (Na, K - wavelength) are not presented in the Materials and methods section. Please give details regarding the quality assurance of the chemical analyzes; the limit of detection should also be included. The tea samples were investigated as infusions, while the used reference material is a solid sample (tea leaves).; no details regarding it preparation, analysis, etc. are presented.

The first time when the authors use an abbreviation in the text, the authors should present both the spelled-out version and the short form. After the definition of an abbreviation, they should use only the abbreviation. Example: chemical symbol of elements.

The “Results and discussion” section must be considerably improved/ more technically presented. In current state, the level of sections is weak and confusing. The manuscript seems to be only an enumeration of information/ obtained results. This issue should be corrected by connecting/ merging the information presented. Furthermore, overall, the presentation of the obtained results appears too long.

The conclusion section should summarize the manuscript's results, discuss unclear data and endorse further research. Furthermore, an effective conclusion should offer closure for a paper, leaving the reader feeling satisfied that the concepts have been fully clarified.

I consider that the article requires carefully, extensive revision and resubmitted. If the manuscript will not be considerable improved, I will not recommend its publication.

Reviewer 3 Report

This research work deals with an intensive study of the contents in phenolics and minerals in many teas.

The manuscript is well written but some typing mistakes need to be corrected, ex. "et al." use the italic style.

This work could be published in selected journal with minor revision: I suggest to improve the description of the purpose of the work and its usefulness for scientific world and better describe the conclusions.

Round 2

Reviewer 2 Report

Generally, I am satisfied with the provided corrections. Indeed, there are impressive amount of results.  However, the English must be improved. Also, the conclusions section needs to be revised, as it is too lengthy, containing unnecessary information. In this regard, this section should be more concise and to the point, while keeping only selected and highlighted main findings.
